# γδ T Cells: Game Changers in Immune Cell Therapy for Cancer

**DOI:** 10.3390/cancers17071063

**Published:** 2025-03-21

**Authors:** Nabil Subhi-Issa, Daniel Tovar Manzano, Alejandro Pereiro Rodriguez, Silvia Sanchez Ramon, Pedro Perez Segura, Alberto Ocaña

**Affiliations:** 1Department of Immunology, Hospital Clínico San Carlos, IdISSC, 28040 Madrid, Spain; 2Department of Immunology, Ophthalmology, and ORL, School of Medicine, Complutense University, 28040 Madrid, Spain; 3Department of Oncology, Hospital Clínico San Carlos, IdISSC, 28040 Madrid, Spainalberto.ocana@salud.madrid.org (A.O.)

**Keywords:** γδ T cells, cancer immunotherapy, cellular expansion protocols, adoptive cell therapy, tumor microenvironment

## Abstract

Gamma delta (γδ) T cells are a distinct subset of T lymphocytes with unique properties that make them promising candidates for cancer immunotherapy. Unlike conventional T cells, γδ T cells recognize tumor antigens in an MHC-independent manner, allowing for broad tumor targeting. They exhibit strong cytotoxic activity and play a key role in shaping the tumor microenvironment. This review focuses on optimizing in vitro expansion protocols to enhance their clinical applicability, detailing strategies such as activation methods, co-culture systems, and cytokine supplementation. Additionally, we discuss recent advancements in γδ T cell research and ongoing clinical applications, while identifying challenges that need further investigation.

## 1. Overview of γδ T Cells in the Immune System

T cells are a critical component of the adaptive immune system, with their primary subtypes—αβ T cells and γδ T cells—defined by the type of T cell receptor (TCR) expressed on their surfaces. While αβ T cells make up the majority of the T cell population and are well-characterized for their MHC-restricted antigen recognition, γδ T cells are a unique and relatively rare subset in humans, typically comprising 1–10% of circulating T cells but found in higher proportions within other tissues such as the gut epithelium, dermis, spleen, and liver. These tissue-resident γδ T cells play a critical role in maintaining local immune surveillance and homeostasis, often exhibiting distinct phenotypes and functional capacities compared to their circulating counterparts [1,2,3,4]. γδ T cells possess distinctive properties that allow them to play a bridge role between innate and adaptive immunity. Unlike αβ T cells, they recognize antigens in an MHC-independent manner, allowing them to respond more rapidly to stressed or transformed cells, including those in the tumor microenvironment (TME). This ability makes γδ T cells especially attractive as candidates for cancer immunotherapy, as they can recognize and attack malignant cells without requiring prior antigen processing and presentation by MHC molecules [5].

In the setting of cancer, γδ T cells exhibit several advantages over αβ T cells for their therapeutic use in humans. Their TCRs, encoded by the γ and δ chains in humans, recognize a range of stress-induced antigens commonly upregulated on tumor cells. Furthermore, γδ T cells have shown the ability to exert both direct cytotoxicity against cancer cells and indirect antitumor effects through cytokine release and crosstalk with other immune cells. Thus, understanding and harnessing these cells is a rapidly growing area of interest in cancer immunology [6].

Over the past few years, several comprehensive reviews have explored different aspects of γδ T cells in immunotherapy, providing valuable insights into their biology, therapeutic applications, and challenges. For instance, a recent review discusses emerging therapeutic strategies involving γδ T cells, including combination therapies and genetic modifications to enhance their efficacy [7]. Another review in Signal Transduction and Targeted Therapy provides an in-depth analysis of γδ T cell origin, subset differentiation, and their role in diseases, including cancer [8]. Additionally, a high-impact review outlines the latest advancements in γδ T cell-based cancer immunotherapy, with a particular focus on their unique tumor-targeting mechanisms and potential clinical applications [6]. Furthermore, a review published in Nature Reviews Clinical Oncology provides a broad overview of γδ T cell-based cancer immunotherapies, covering their unique biology, clinical progress, and limitations [5]. This review also highlights the translational challenges, including TME-induced immunosuppression and persistence of γδ T cells post infusion, which are critical considerations for optimizing in vitro expansion protocols.

Here, we comprehensively review the current data on the potential of γδ T cells for use in cancer immunotherapy, along with the latest advancements in protocols for γδ T cell expansion, with an emphasis on understanding how these protocols can be optimized to enhance antitumor functionality. Through this review, we seek to summarize the current state of γδ T cell expansion protocols, provide insights into their limitations and challenges, and discuss the potential of combining γδ T cell therapy with other immunotherapeutic approaches for enhanced cancer treatment. Through these updated data, we present a hypothesis for potential optimized therapeutic use.

## 2. Differences Between γδ T Cells and αβ T Cells

The structural and functional differences between γδ and αβ T cells underscore the unique therapeutic potential of γδ T cells. γδ T cells do not possess entirely unique functions; however, they can provide distinctive combinations of functional abilities, including cytolysis, the induction of IgE, antigen presentation, and the production of growth factors [9].

The TCR on αβ T cells consists of α and β chains, while γδ T cells utilize a distinct TCR structure made up of γ and δ chains, enabling them to detect a broader spectrum of antigens, without reliance on MHC [9], which allows them to target a broader range of cancer cells, including those with low mutational burdens or downregulated MHC expression. This unique capability allows them to recognize both external and internal antigens, including those from foreign sources and the body’s own cells. αβ T cells are largely dependent on antigen processing and presentation through MHC molecules, which limits their ability to recognize certain transformed or infected cells. In contrast, γδ T cells recognize non-MHC antigens presented directly on the surface of target cells, a process facilitated by molecules like butyrophilin (BTNs) family proteins [10,11]. This MHC-independent recognition enables γδ T cells to rapidly detect and respond to cellular changes such as infection or malignancy [12]. Additionally, γδ T cells exhibit tissue tropism and can function as both innate and adaptive immune effectors, providing rapid and versatile responses. They also have the potential to be used in allogeneic settings without causing graft-versus-host disease (GvHD) [5,13,14,15].

γδ T cells display surface NK cell receptors (NKRs) that enhance their cytotoxic activity when they engage specific ligands on target cells. Notably, the C-type lectin-like receptor NKG2D interacts with stress-induced MHC class I-related proteins, MICA and MICB [16]. Additionally, like NK cells but unlike αβ T cells, γδ T cells express the functional receptor CD16 (also known as low-affinity IgG Fc receptor III or FcγRIII), which enables them to perform antibody-dependent cellular cytotoxicity (ADCC) against tumor cells marked by antibody-based therapies [17,18].

The innate nature of γδ T cells also allows them to mount immediate responses against malignancies, a key attribute leveraged in developing γδ T cell-based immunotherapies. In contrast, γδ T cells face challenges, such as limited abundance, difficulties in expansion and persistence, and complex biology that can hinder their therapeutic potential. Efforts are ongoing to address these issues through genetic engineering strategies, such as chimeric antigen receptor (CAR) T cell therapy and TCR gene transfer [14].

## 3. Classification and Anti-Tumor Mechanisms of γδ T Cells

γδ T cells are classified based on the variable regions of their TCR γ and δ chains, which define unique subsets with distinct tissue distributions, antigen recognition capacities, and functional roles. In humans, γδ T cells are primarily divided into three subsets based on their TRDV genes: Vδ1^+^, Vδ2^+^, and Vδ3^+^. A comparative overview of γδ T cell subsets and αβ T cells is shown in Table 1. The Vδ2^+^ subset commonly pairs with the Vγ9 TCR, making it the most abundant γδ T cell type in peripheral blood. Vγ9Vδ2 cells emerge during early fetal development, unlike Vδ1^+^ cells, which develop in the thymus a few months post birth [19,20]. This early emergence suggests that Vγ9Vδ2 cells act as a frontline defense component and play a key role in innate immunity [8,21].

Vγ9Vδ2^+^ T cells have the unique ability to recognize phosphoantigens (pAgs) derived from the mevalonate pathway, which is often dysregulated in tumor cells. The most potent activator is hydroxymethyl-but-2-enyl pyrophosphate (HMBPP), an intermediary in the deoxyxylulose (or non-mevalonate) pathway for cholesterol synthesis. Additionally, Vγ9Vδ2^+^ T cells can also be activated by isopentenyl pyrophosphate (IPP), a self-produced pAg that acts as an intermediary in the mevalonate pathway [35,36]. Vγ9Vδ2 cells are also known for their potent cytotoxic functions, rapid cytokine release (e.g., IFN-γ and TNF-α), and possess a strong ability to cross-present antigens, much like dendritic cells, thereby stimulating CD8^+^ αβ T cell responses [9,27,37,38]. Vγ9Vδ2^+^ cells have demonstrated robust anti-tumor activity in preclinical models and early clinical trials, particularly in triple-negative breast cancer (TNBC) and glioblastoma [39,40].

Vδ1^+^ T cells comprise only about one-third of circulating γδ T cells. They are largely tissue-resident and are found in epithelial tissues, including the skin, gut, and lungs, where they contribute to barrier immunity and tissue repair [5]. Vδ1^+^ T cells exhibit a different antigen recognition range than Vγ9Vδ2^+^ cells, responding to stress ligands and other non-peptidic antigens often upregulated in damaged or transformed tissues. The Vδ1 TCR specifically binds to MHC-like proteins within the CD1 family, such as CD1c and CD1d, as well as Annexin A2 and stress-induced ligands such as MICA/B and UL16-binding proteins, which are often upregulated in tumor cells [34,41,42,43,44]. This subset has been associated with improved outcomes in acute myeloid leukemia (AML), particularly in the context of allogeneic hematopoietic stem cell transplantation, where they contribute to graft-versus-leukemia effects without causing graft-versus-host disease [45,46]. NKp46 is highly expressed on Vδ1 T cells localized in the intestinal epithelium, where it characterizes a specific subset with heightened cytotoxic activity against colorectal cancer cells [23]. NKp30 and NKp44 are newly expressed on Vδ1 T cells following TCR stimulation combined with IL-2 or IL-15 in vitro, enhancing their antitumor activity against hematological malignancies and potentially solid tumors [47].

Vδ1^+^ T cells are typically more abundant than Vδ2^+^ T cells within tumor infiltrates, and γδ T cell lines derived from tumor-infiltrating Vδ1^+^ cells have generally demonstrated superior cytotoxic activity against tumors in in vitro assays compared to Vδ2^+^ TIL-derived lines [48,49]. The distinct ligand recognition profiles of Vδ1^+^ and Vδ2^+^ T cells suggest that each subset plays a unique, non-overlapping role in immune surveillance, which could offer unique advantages depending on the tumor type and therapeutic context.

Although less studied, other subsets have been described to contribute to tissue-specific immune responses and may play distinct roles in immune surveillance, infection control, and tumor recognition. Vδ3^+^ T cells make up a very small fraction of peripheral blood lymphocytes but are found in higher concentrations in the liver and gut. Their numbers increase in patients with cytomegalovirus activation and B cell chronic lymphocytic leukemia [50].

γδ T cells are increasingly recognized for their powerful antitumor capabilities, which are mediated through a variety of mechanisms (Figure 1):

Direct Cytotoxicity: Both Vγ9Vδ2 and Vδ1 subsets demonstrate direct cytotoxic activity against tumor cells. Through engagement of the TCR with stress ligands or non-peptide antigens on malignant cells, γδ T cells can rapidly mobilize to release perforin and granzymes, inducing apoptosis in targeted cells [51]. Human γδ T cells express various surface molecules that recognize tumor cells and deliver further stimulatory signals. Among these are natural killer receptors, including NKG2D, DNAX Accessory Molecule-1 (DNAM-1), NKp30, NKp44, and NKp46, which allow γδ T cells to detect and respond to malignant transformations effectively [52].

Antibody-Dependent Cellular Cytotoxicity (ADCC): γδ T cells, particularly those expressing CD16 (FcγRIII), can mediate ADCC by recognizing antibodies bound to tumor cells [17]. This function is valuable in combination with monoclonal antibody therapies, such as rituximab and trastuzumab, which tag cancer cells for destruction [27].

Cytokine and Chemokine Release: γδ T cells are notable for their rapid and robust production of pro-inflammatory cytokines, such as IFN-γ, TNF-α, and IL-17, upon activation [53]. IFN-γ and TNF-α contribute to the anti-TME by promoting tumor cell apoptosis, enhancing antigen presentation, and recruiting other immune cells to the tumor site [54]. IL-17, while controversial in its role, can have both pro-tumor and anti-tumor effects depending on the context [55]. These cytokines position γδ T cells as critical modulators within the TME, with potential to orchestrate a broader immune response against cancer.

TRAIL and FasL expression: Beyond the perforin–granzyme pathway, γδ T cells can also eliminate tumor cells through the expression of TNF-related apoptosis-inducing ligand (TRAIL) and Fas ligand (FasL), which bind to their corresponding receptors on tumor cells to trigger apoptosis [28,56].

While γδ T cells are widely recognized for their antitumor properties, certain subsets and functional profiles have been implicated in tumor promotion under specific conditions. Understanding these tumor-promoting capabilities is essential to fully harness the therapeutic potential of γδ T cells while mitigating unintended effects.

One of the most prominent tumor-promoting subsets of γδ T cells are those that produce interleukin-17 (IL-17). These IL-17-producing γδ T cells (also known as γδT17 cells) contribute to tumor progression by fostering a pro-inflammatory microenvironment. IL-17 has been shown to enhance the recruitment of myeloid-derived suppressor cells (MDSCs) and neutrophils, both of which suppress antitumor immune responses [57,58]. Additionally, IL-17 promotes angiogenesis both directly and indirectly. It can directly signal endothelial cells to stimulate blood vessel formation [59], and, indirectly, it induces macrophages to produce angiogenic factors, such as vascular endothelial growth factor (VEGF) [60].

In humans, while Vγ9Vδ2^+^ T cells can secrete IL-17 in response to stimulation with antigens and cytokines such as IL-1β, IL-6, IL-23, and TGFβ [61], IL-17 is predominantly produced by Vδ1^+^ T cells [62]. Tissue-resident innate γδ T cells, which are more inclined to produce IL-17, differ from circulating γδ T cells that preferentially produce IFNγ. Therefore, tissue-resident Vδ1+ T cells, rather than Vδ2^+^ T cells, are likely the primary sources of IL-17. However, the cytokine profiles of γδ T cell subsets are highly context-dependent. For instance, in breast cancer, tissue-resident Vδ1+ T cells tend to exhibit cytotoxic activity and produce IFNγ rather than IL-17 [63]. Consequently, attributing cytokine production solely based on TCR chains may oversimplify the complex biology of γδ T cells.

Interestingly, recent findings have identified neutrophil-derived reactive oxygen species (ROS) as important regulators of pro-tumoral γδT17 cells [64]. These cells express particularly low levels of the antioxidant glutathione, making them highly sensitive to oxidative stress. This discovery opens new avenues for clinical translation by targeting ROS modulation to mitigate the tumor-promoting activities of γδT17 cells. Optimizing γδ T cell-based therapies requires careful consideration of these dynamics, focusing on suppressing IL-17-producing subsets while enhancing the antitumor activity of cytotoxic γδ T cells to improve therapeutic outcomes.

Despite these advantages, their therapeutic applications in cancer treatment remain challenging due to their heterogeneity, potential tumor-promoting subsets, and difficulties in achieving sustained antitumor responses. The following section explores the clinical applications of γδ T cells, discussing their use in both hematologic malignancies and solid tumors, highlighting both their successes and the remaining hurdles in translating these findings into effective cancer immunotherapies.

## 4. γδ T Cell-Based Immunotherapies in Cancer: Clinical Applications and Challenges

### 4.1. Clinical Outcomes of γδ T Cell-Based Therapies

Over the past decade, γδ T cell-based therapies have gained traction in cancer immunotherapy due to their MHC-independent tumor recognition and potent cytotoxicity. While preclinical studies have shown promising results, the clinical success of γδ T cell therapies remains variable, largely due to challenges in ex vivo expansion protocols, persistence in vivo, and TME resistance.

Ex vivo expansion of autologous γδ T cells using aminobisphosphonates or synthetic pAg, followed by adoptive cell transfer (ACT), has been evaluated across various malignancies, including multiple myeloma, renal cell carcinoma, non-small-cell lung cancer, gastric cancer, hepatocellular carcinoma, melanoma, ovarian cancer, colon cancer, and pancreatic cancer [65,66,67,68,69,70,71,72,73,74,75]. While these strategies have demonstrated a favorable safety profile, their clinical efficacy has generally been limited, with responses occurring infrequently and lacking durability, despite occasional notable outcomes.

For instance, in a Phase I study of γδ T cell infusion for advanced esophageal cancer, the median overall survival (OS) was 5.7 months, with a median progression-free survival (PFS) of 2.4 months. Similarly, a non-small-cell lung cancer (NSCLC) trial reported an ORR of only 4% with γδ T cell-based therapy despite a median OS of 418 days. In Table 2, a summary of notable γδ T cell clinical trials and their reported outcomes is shown. The overall modest antitumor effects observed with systemic γδ T cell activation via aminobisphosphonates or synthetic pAg, as well as with autologous γδ T cell transfer, have hindered further advancement of γδ T cell-based cancer immunotherapies.

Several clinical studies have demonstrated that γδ T cells play an essential role in the graft-versus-leukemia (GvL) effect following allogeneic hematopoietic stem cell transplantation (allo-HSCT). Lamb et al. reported that leukemia patients who received HLA-mismatched allo-HSCT-depleted αβ T cells found that a higher proportion of γδ T cells (≥10%) in the first 6 months post transplantation correlated with better disease-free survival (DFS) and lower relapse rates [77]. These Vδ1^+^CD69^+^ γδ T cells were cytotoxic against leukemia cells in vitro, and in a follow-up study involving 153 acute leukemia patients, they confirmed an overall survival (OS) benefit with γδ T cell expansion [78]. Similar positive correlations between γδ T cell expansion and patient outcomes were observed in pediatric leukemia studies [79].

However, despite these promising findings, autologous γδ T cells expanded ex vivo have shown limited efficacy in leukemia treatment. In a prospective clinical trial by Kunzmann et al., AML patients treated with zoledronic acid (ZOL) and IL-2 experienced an increase in γδ T cells and IFN-γ production, but only two out of eight patients (25%) achieved partial remission (PR) [80]. This suggests that while γδ T cells can exert a GvL effect, their autologous expansion alone is often insufficient for robust clinical responses.

One approach to overcome these limitations is the use of allogeneic γδ T cell therapy. In a subsequent study, four patients with refractory hematologic malignancies received haploidentical family donor-derived γδ T cells after CD4/CD8 depletion and ZOL infusions [81]. Three of the four patients (75%) achieved complete remission (CR) without developing GvHD, indicating that allogeneic γδ T cell therapy may be a safer and more effective approach than autologous γδ T cell infusion.

In a study by Wilhelm et al., 19 patients with relapsed or refractory low-grade non-Hodgkin lymphoma (NHL) or MM received ZOL and IL-2 infusions to stimulate γδ T cell expansion in vivo. However, only three patients (33%) who achieved significant γδ T cell expansion experienced partial remission (PR), suggesting that γδ T cell-based therapies require additional optimization for improved efficacy [82].

Similarly, Abe et al. conducted the first adoptive γδ T cell therapy trial for MM by expanding Vγ9Vδ2 T cells in vitro with ZOL before reinfusion into patients [70]. Unfortunately, none of the nine MM patients achieved clinical benefit, underscoring the need for improved expansion protocols to enhance therapeutic efficacy.

Understanding past clinical outcomes provides valuable insights into designing more effective ongoing and future trials, guiding improvements in γδ T cell therapy.

### 4.2. Current Clinical Trials and Combination Strategies

Several small-scale clinical trials indicate that γδ T cell-based therapies are safe and may offer clinical benefits. However, the objective response rates have been low, indicating the need for larger and well-controlled trials to better understand and harness their therapeutic potential (reviewed in [83,84]). There have been promising results in glioblastoma [85], TNBC in preclinical trials [86], and AML [45], showing a good safety profile in all of them. Table 3 provides an overview of ongoing clinical trials.

Survival endpoints, such as overall survival (OS), progression-free survival (PFS), and relapse-free survival (RFS), are pivotal in trials like NCT05015426 and NCT03533816, particularly in high-risk hematologic malignancies. For instance, NCT05015426 emphasizes leukemia-free survival (LFS) and OS in post-alloHCT patients with acute myeloid leukemia (AML). Similarly, NCT03533816 includes RFS and OS as critical measures to assess the longevity of treatment effects following γδ T cell infusion after stem cell transplantation. Trials such as NCT04165941, targeting glioblastoma multiforme (GBM), incorporate OS and PFS as benchmarks to compare the efficacy of γδ T cell therapy against standard treatment protocols.

These survival endpoints provide essential insights into the durability of γδ T cell-mediated antitumor responses. However, extending these analyses beyond short-term results to long-term survival is critical, as shown in NCT05358808, which evaluates responses longer than one year.

Quality of life (QoL) remains an underexplored aspect of γδ T cell therapy trials. Among the reviewed trials, only NCT04764513 explicitly integrates QoL endpoints using tools such as the EORTC QLQ-C30, which assess physical, emotional, and social well-being. This is a critical addition, as QoL measures provide insights into the holistic benefits of γδ T cell therapies, especially when compared to CAR-T cell therapies, which are often associated with severe toxicities like cytokine release syndrome (CRS). Incorporating QoL assessments in trials like NCT04165941 and NCT05400603, which involve complex regimens or pediatric populations, would offer valuable data on the impact of γδ T cell therapies on daily functioning and long-term patient satisfaction.

Predictive biomarkers are increasingly being explored to enhance patient selection and optimize therapeutic outcomes. Trials NCT04735471 monitors the pharmacokinetics of CAR-engineered γδ T cells, offering insights into their in vivo dynamics and correlating these with clinical responses.

NCT04165941 also incorporates immunophenotyping and cytokine analyses to identify response-related biomarkers in patients with glioblastoma [87].

Future trials should prioritize the inclusion of biomarkers to guide patient stratification and optimize efficacy.

While the number of cells infused varies across studies (from 2 × 10^6^ cells/kg to 5 × 10^6^ cells/kg), this parameter is crucial for ensuring effective dosing and maximizing therapeutic impact. Supplementation with cytokines, particularly IL-2, has been employed to support γδ T cell persistence and functionality post infusion (NCT03533816). High-dose IL-2 can lead to significant toxicity, manifesting as hyperpyrexia, capillary leakage, and hypotension. While low-dose IL-2 is better tolerated, it may counteract its intended effects in cancer immunotherapy. For instance, although it promotes the effector memory phenotype in T cells after TCR activation, it also increases the number of circulating regulatory T cells (Tregs), thereby inducing a strong immunosuppressive response [88]. However, the potential use of IL-15 as an alternative to IL-2 warrants exploration, given its ability to enhance memory-like properties in γδ T cells, prolong cell survival, and reduce the risk of activation-induced cell death.

Allogeneic γδ T cell-based therapies offer the potential for off-the-shelf immunotherapies, reducing the time and cost associated with autologous approaches. While Vγ9Vδ2 T cells have been extensively studied in allogeneic settings, Vδ1^+^ T cells present distinct advantages that make them particularly suited for this application such as superior persistence [89], tissue tropism [90], and possibly broader tumor-associated ligand recognition.

The integration of γδ T cell therapy with other immunotherapeutic modalities offers promising avenues to enhance antitumor efficacy, address resistance mechanisms, and improve patient outcomes.

Pairing γδ T cells with immune checkpoint inhibitors like anti-PD1, represent an intriguing avenue for boosting antitumor responses. These combinations could address challenges related to tumor-induced immune suppression and enhance γδ T cell-mediated cytotoxicity. Blocking PD-1 with antibodies like pembrolizumab does not substantially affect the cytotoxic activity of ZOL-activated Vδ2 T cells [91] but significantly boosts their IFN-γ production [92]. Given the PD-1 expression status on γδ T cells, combining pembrolizumab with γδ T cell therapy could be a viable strategy, akin to its recent successful application in allogeneic NK cell therapy [93]. NKG2A has emerged as a novel checkpoint inhibitor, and the humanized anti-NKG2A monoclonal antibody, monalizumab, has been shown to enhance antitumor immunity by boosting the activity of CD8 T cells and NK cells [94,95]. Although these studies have not specifically explored its impact on γδ T cells, it would be valuable to examine how monalizumab influences the activation and effector functions of NKG2A-expressing γδ T cells.

As far as we know, no studies have yet been published directly combining αβ CAR-T cell therapy with γδ T cell immunotherapy, but the potential rationale for such a combination lies in the complementary mechanisms of these two cell types. CAR-T cells, engineered to target specific tumor antigens, are highly effective in targeting antigen-expressing tumor cells. γδ T cells, with their innate ability to recognize stress ligands and perform MHC-independent cytotoxicity, could theoretically complement CAR-T cells by targeting antigen-negative tumor cells and enhancing immune responses in the TME through cytokine secretion.

However, CAR-T cell therapy is frequently associated with significant side effects, such as cytokine release syndrome (CRS) and immune effector cell-associated neurotoxicity syndrome (ICANS). Introducing γδ T cells into this therapeutic landscape could potentially amplify these risks due to their ability to produce inflammatory cytokines and mediate strong cytotoxic responses. The potential for exacerbating these severe toxicities highlights a critical limitation of such a combination, making it imperative to resolve these safety concerns before clinical exploration of αβ CAR-T and γδ T cell combinations.

Further research is needed to evaluate whether this theoretical synergy can be safely harnessed. This might include engineering γδ T cells or CAR-T cells with enhanced safety controls, such as regulatory switches, or optimizing combination strategies to limit overactivation. Until these challenges are addressed, combining CAR-T cells and γδ T cells remains a conceptually intriguing but currently impractical approach.

Bispecific antibodies offer a promising approach to enhance γδ T cell-mediated immunotherapy by bridging these innate T cells with tumor-associated antigens, thereby improving their cytotoxic targeting and expanding their therapeutic potential. One example of this strategy involves a bispecific antibody targeting the HER2 antigen and the Vγ9 TCR. This construct has demonstrated the ability to amplify Vγ9Vδ2 T cell cytotoxicity against HER2-positive pancreatic cancer cells both in vitro and in a subcutaneous pancreatic ductal adenocarcinoma (PDAC) mouse model [96]. Similarly, bispecific antibodies targeting CD40 or CD1d have shown success in preclinical models of hematological malignancies, such as chronic lymphocytic leukemia (CLL) and multiple myeloma (MM) [97,98]. For example, a CD40-specific bispecific antibody effectively redirected Vγ9Vδ2 T cells to kill tumor cells while simultaneously blocking the pro-survival signaling of CD40, resulting in prolonged survival in mouse models of CLL. Additionally, a CD1d-specific engager not only enhanced Vγ9Vδ2 T cell-mediated killing of CLL and MM cells but also induced pro-inflammatory cytokine secretion and dendritic cell maturation, suggesting a dual role in both direct tumor cell lysis and immune system activation. Another innovative development includes gamma delta TCR anti-CD3 bispecific molecules (GABs), which target multiple tumor types by linking Vγ9Vδ2 TCRs to CD3 on T cells. These molecules have demonstrated efficacy in enhancing tumor infiltration, cytokine secretion, and tumor lysis in preclinical models [99]. Finally, tribody constructs targeting CD16 and tumor antigens, such as CD19, have shown robust engagement of both NK and γδ T cells, leading to enhanced antibody-dependent cellular cytotoxicity (ADCC) compared to monoclonal antibodies like rituximab [100]. These tribody designs present a cost-effective and scalable alternative to adoptive cell therapy, broadening the clinical applicability of γδ T cell-mediated cancer immunotherapy.

While the protocols for γδ T cell expansion have shown promising results, several challenges remain. One significant issue is the limited yield of certain subsets, particularly Vδ1^+^ T cells, which have demonstrated potential in targeting solid tumors but are more challenging to expand compared to Vγ9Vδ2 cells. Unlike Vγ9Vδ2 T cells, which can be readily expanded using pAgs, Vδ1^+^ T cells do not respond to phosphoantigen stimulation. This fundamental difference in antigen recognition necessitates alternative expansion strategies for Vδ1-based immunotherapies. Most in vitro expansion protocols for Vδ1^+^ T cells rely on mitogenic plant lectins, such as phytohemagglutinin (PHA) and concanavalin A (ConA) [89]. These lectins induce robust TCR-mediated activation and proliferation, allowing for the large-scale expansion of these cells. However, despite their effectiveness, these reagents pose significant clinical limitations, such as safety concerns, lack of specificity and limited control over functional phenotypes.

The advancement of Vδ1 ACT has been facilitated by clinical-grade protocols that enable substantial expansion of these cells from both tissue and blood sources. Notably, the Delta One T (DOT) protocol achieves over a 2000-fold cell expansion within 2 to 3 weeks, while simultaneously promoting the upregulation of natural killer receptors (NKRs), which enhance the targeting of tumor cells [90]. Developing targeted protocols to efficiently expand Vδ1 T cells without compromising their antitumor function is an area of active research.

Another challenge is maintaining the functional integrity of γδ T cells during expansion. Sustaining their cytotoxic function and preventing phenotypic drift during culture are critical hurdles. Strategies to reduce cell exhaustion and maintain key effector functions throughout the culture period, such as optimizing cytokine combinations and controlling the expansion duration, are currently under investigation. In fact, different markers expressed by γδ T cells have been associated with favorable prognosis in some human tumors, such as CD69, IFN-γ or NKp46.

Finally, the successful translation of expanded γδ T cells into clinical use hinges on the optimization of protocols that produce clinical-grade cells meeting GMP standards. Moreover, combining expanded γδ T cells with other immunotherapies, such as checkpoint inhibitors or CAR T cells, represents an exciting opportunity to enhance therapeutic efficacy.

Given the barriers posed by the tumor microenvironment and the variable success of γδ T cells in clinical settings, effective in vitro expansion protocols are critical to generating large numbers of functionally competent γδ T cells for adoptive therapy. However, the optimal conditions for γδ T cell expansion remain an area of active investigation, with various approaches being explored, including different activation stimuli, co-culture systems, and cytokine combinations.

## 5. Optimization of γδ T Cell Expansion for Therapeutic Applications

γδ T cells constitute only 1–10% of the circulating T cell population, which poses a significant challenge for their use in immunotherapy. This scarcity necessitates efficient methods for their isolation and expansion to achieve clinically relevant cell numbers [14]. Expanding γδ T cells in vitro has emerged as a critical step in developing γδ T cell-based immunotherapies for cancer and infectious diseases. γδ T cells exhibit unique antigen recognition properties, including the ability to recognize non-peptidic antigens in an MHC-independent manner, providing a promising avenue for therapies targeting a wide range of tumors without the limitations of HLA matching. The main goal of in vitro expansion is to generate enough functional γδ T cells that retain their cytotoxic activity, memory characteristics, and ability to home to tumor sites when infused into patients. However, expansion protocols must carefully balance proliferation with the preservation of these effector functions and phenotype stability. Additionally, maintaining the persistence of these cells in vivo post infusion is another hurdle, as they often exhibit limited survival and activity in the TME [101].

### 5.1. Activation and Stimulation Approaches

The initial step in γδ T cell expansion typically involves robust activation, which is crucial for cell proliferation. A variety of stimulants are employed to activate these cells. pAg play a significant role in stimulating Vγ9Vδ2 T cells through their TCR, resulting in rapid proliferation and activation. The addition of aminobisphosphonates, such as zoledronic acid, or synthetic pAg, like bromohydrin pyrophosphate (BrHPP), enhances the expansion and activation of γδ T cells, although through different pathways. Aminobisphosphonates act by inhibiting farnesyl pyrophosphate synthase in the mevalonate pathway, which leads to intracellular accumulation of the pAg IPP, thereby boosting γδ T cell activation [36,102,103]. However, aminobisphosphonates were unable to trigger proliferative responses in isolated γδ T cell clones; they require the presence of additional cells, such as monocytes, in the culture to effectively induce γδ T cell expansion [104]. Additionally, BrHPP functions as a pAg agonist. When γδ T cells are exposed to BrHPP, it stimulates the release of TNF-α and IFN-γ, suggesting that this ligand activates the complete spectrum of γδ T cell effector responses [105,106].

Monoclonal antibodies also contribute significantly to γδ T cell expansion by providing essential co-stimulatory signals. For example, anti-CD3 monoclonal antibodies (mAbs) are often combined with pAg or cytokines to offer a secondary activation signal that enhances γδ T cell proliferation. The binding of anti-CD3 to the CD3 molecule on T cells, a critical co-receptor for TCR signaling, amplifies the activation process [107]. Another promising approach for γδ T cell activation involves targeting butyrophilin (BTN) family molecules, specifically butyrophilin 3A (BTN3A). The development of mouse anti-human CD277 antibodies has been instrumental in advancing our understanding of Vγ9Vδ2 T-cell activation mechanisms and holds significant promise for therapeutic applications [10,108]. These antibodies activate γδ T cells through pathways that operate downstream of, and independently from, IPP, offering an alternative to traditional pAg stimulation. Notably, the activating anti-CD277 antibody clone 20.1 demonstrates similar, though not identical, stimulatory effects to pAg activation, potentially targeting specific Vγ9Vδ2 T-cell subsets with particular TCR complementarity-determining region sequences [109]. This property suggests a tailored activation approach that could surpass aminobisphosphonates and other metabolic sensitizers, especially in targeting cells that show resistance to drug internalization or reduced activity in the mevalonate pathway.

In addition to TCR-mediated activation, γδ T cells can also be stimulated through activating receptors, particularly those commonly associated with NK cell-like functions, such as NKG2D. The activation pathway is independent of TCR signaling, allowing γδ T cells to exert rapid effector responses against a broad spectrum of malignancies, particularly those with high levels of NKG2D ligands. Incorporating NKG2D ligands in vitro or using recombinant proteins that engage NKG2D can significantly enhance γδ T cell expansion and functional potency, by amplifying T-cell cytokine production, proliferation, and cytotoxicity in vitro [26,110,111].

Cytokines are another key component in the expansion of γδ T cells, influencing their survival and proliferation. Interleukin-2 (IL-2) is routinely used in expansion protocols due to its ability to support γδ T cell growth and maintain cytotoxic function [112,113]. Other cytokines, including IL-15 and IL-21, are increasingly utilized to improve both the activation profile and substantially enhance the antitumor activity of γδ T cells. Specifically, IL-15 has been shown to promote memory-like characteristics in γδ T cells [114], while IL-21 enhances their cytotoxic capabilities against tumor cells [40,115]. Nevertheless, the effectiveness of IL-21 was constrained, as it also elevated the expression of the checkpoint molecule Tim-3. This limitation was particularly notable in AML patients, who exhibited lower levels of IL-21R and higher Tim-3 expression [116].

One notable example involved the engineering of γδ T cells to secrete synthetic tumor-targeting opsonins and a mitogenic IL-15Rα-IL-15 fusion protein (stIL15) in osteosarcoma. This approach showed enhanced cytotoxicity and promoted bystander activity in other lymphoid and myeloid cells. Specifically, the secretion of stIL-15 by these engineered γδ T cells (stIL15-OPS-γδ T cells) obviated the need for exogenous cytokine supplementation and mediated the activation of bystander NK cells at TME. These engineered cells demonstrated superior in vivo control of subcutaneous tumors and persistence in the blood compared to unmodified γδ T cells [117].

To ensure effective γδ T cell expansion, activation strategies may be complemented with appropriate co-culture systems that provide the necessary support for cell proliferation and function.

### 5.2. Co-Culture Systems and Feeder Cells

Co-culture systems involving feeder cells or accessory cells can provide additional signals necessary for γδ T cell expansion. Peripheral blood mononuclear cells (PBMCs) are commonly used as feeder cells in αβ T cells expansion protocols, as they supply natural cytokines and co-stimulatory signals that facilitate T cell activation and proliferation [118]. To prevent overgrowth while still supporting T cell expansion, PBMCs are often irradiated [119,120,121]. Notably, the use of PBMC feeders has not been widely adopted in protocols specifically designed for expanding γδ T cells.

Using PBMCs as feeder cells in T cell expansion protocols presents several significant limitations. First, PBMC-based expansion requires large quantities of these cells to sustain optimal growth and activation of T cells over extended culture periods. This demand for high cell numbers can complicate the scalability of protocols and is particularly challenging in clinical settings, where patient-specific PBMCs may be in limited supply [121]. Additionally, the mechanisms by which PBMCs support T cell expansion are not fully understood, with a lack of precise knowledge regarding the specific markers and ligands on PBMCs that most effectively enhance T cell proliferation and activation. This gap in understanding limits the ability to optimize PBMC-based feeder systems and may lead to inconsistent results across different protocols. Another key limitation is the inherent heterogeneity in PBMC composition between patients or donors, which introduces variability in the quality and efficacy of T cell expansion. Factors such as age, immune status, and underlying health conditions can affect the cell populations within PBMCs, resulting in inconsistent activation and expansion responses when used as feeders. Consequently, these challenges underscore the need for more standardized and predictable feeder systems that can reliably support robust and reproducible T cell expansion across diverse patient populations.

Artificial antigen-presenting cells (aAPCs) have been developed to provide a more defined and reproducible method for stimulating γδ T cells. These aAPCs express ligands for CD3 and co-stimulatory molecules (such as CD137L and CD80), enabling them to effectively promote γδ T cell proliferation while reducing dependence on PBMCs-derived feeder cells, thus enhancing the scalability of expansion protocols [121,122,123,124].

### 5.3. Expansion Protocol Optimization

Optimizing the protocols for γδ T cell expansion for clinical applications requires careful adjustments of several parameters. The duration of the expansion period is an important consideration. γδ T cell cultures are typically expanded over a span of 10 to 14 days, with monitoring of cell phenotype and cytotoxicity. Longer culture periods have so far not been included in clinical studies [67,70,71,72,73,74,81,125,126,127]. Therefore, adjusting cytokine concentrations and periodically replenishing culture media are essential strategies to mitigate exhaustion and maintain functional integrity throughout the expansion process. Figure 2 illustrates the process of expansion from patient-derived cells to reinfusion.

### 5.4. Overcoming T Cell Exhaustion and Enhancing γδ T Cell Persistence in the Tumor Microenvironment

The TME presents significant challenges to γδ T cell-based therapies, including immune suppression, metabolic stress, and induction of exhaustion. Addressing these barriers is essential to improving therapeutic outcomes. One promising approach is cytokine supplementation, particularly with IL-15, which enhances γδ T cell persistence and supports their cytotoxic function by promoting a memory-like phenotype [114,127,128]. In other adoptive cell therapies, such as those utilizing NK cells or CD8+ T cells, IL-15 has demonstrated its ability to synergize with these effector cells, leading to enhanced antitumor activity. In preclinical models, IL-15 has improved tumor regression and survival outcomes by promoting the infiltration and activation of cytotoxic lymphocytes within tumors. IL-36γ stimulates IFN-γ production in CD8+ T cells, NK cells, and γδ T cells, reshaping the TME to support cancer elimination and demonstrating potent antitumor activity [129]. We hypothesize that combining γδ T cell therapy with engineered cytokine delivery systems can optimize antitumor responses by enhancing tumor infiltration, cytotoxic activity, and local immune modulation, thereby improving the effectiveness and safety of cancer immunotherapy.

Another strategy involves the use of immune checkpoint inhibitors like PD-1 and CTLA-4 blockers, or the integration γδ T cells with engineered resistance to inhibitory pathways, such as PD-1 or CTLA-4, to further enhance their functionality and reverse exhaustion in the suppressive conditions of the TME.

Beyond genetic modifications and cytokine supplementation, combining γδ T cells with other immune effectors, such as natural killer (NK) cells or αβ T cells, leverages the complementary strengths of these immune subsets. These combinations can amplify antitumor responses by integrating the non-MHC-restricted cytotoxicity of γδ T cells with the adaptive immunity mediated by αβ T cells [130,131].

Numerous γδ T cell expansion protocols have been developed, each with distinct advantages and limitations regarding yield, functionality, cost-effectiveness, and clinical scalability. A comparative analysis of these approaches is essential to identify the most efficient, reproducible, and clinically translatable methods for large-scale γδ T cell manufacturing.

## 6. Characterization and Comparative Analysis of γδ T Cell Expansion Protocols

The characterization of expanded γδ T cells is essential for assessing their therapeutic potential and optimizing expansion protocols for clinical applications. A variety of surface markers and functional molecules are evaluated to determine the quality, activation state, and cytotoxic potential of these cells after ex vivo expansion. Studies have highlighted several key markers associated with the successful proliferation and function of γδ T cells in immunotherapy settings.

Markers such as PD-1, CTLA-4, Eomes, T-bet, and CD69 have been closely examined, with lower expression of these molecules correlating with enhanced γδ T cell expansion [132]. Reduced levels of inhibitory markers like PD-1 and CTLA-4 are particularly advantageous, as they may indicate a less exhausted phenotype and, therefore, a greater capacity for proliferation and function. Functional cytokine production, specifically IFN-γ, is another critical indicator of effective expansion; higher IFN-γ levels have been associated with improved γδ T cell activation and cytotoxicity [132].

Additionally, CD137L expression on artificial antigen-presenting cells (aAPCs) has been identified as essential for promoting robust γδ T cell proliferation, reinforcing the importance of engineered support cells in expansion protocols [24]. The transcription factor T-bet, when expressed at higher levels, is associated with increased cytotoxicity and proliferation of γδ T cells, particularly when expanded with IL-15, further highlighting the functional versatility of these cells in different cytokine environments [133]. Additionally, Vγ9Vδ2 T cells expanded with IL-2 and IL-15 exhibited enhanced cytotoxicity even in hypoxic conditions. Lastly, markers such as CD25, CD45RO, HLA-DR, CD8, CD16, CD95, and NKG2D are commonly expressed on expanded γδ T cells and are indicative of their activation status and cytotoxic potential, underscoring their readiness for therapeutic use [134,135].

The effective use of γδ T cells in cancer immunotherapy requires the development of robust expansion protocols that are not only scalable and cost-effective but also capable of producing high-quality, functional cells for clinical application. Each method for γδ T cell expansion comes with unique advantages and obstacles, particularly in terms of cost-effectiveness, scalability, and clinical translation.

Feeder cell-based systems, using irradiated PBMCs, are one of the most established approaches. These cells provide natural co-stimulatory signals and cytokines that enhance γδ T cell activation and proliferation. However, feeder-based systems are labor-intensive, requiring large quantities of feeder cells and precise quality control to ensure consistency. The heterogeneity of PBMCs between donors introduces variability, which complicates standardization for clinical-grade production. Additionally, the reliance on patient- or donor-derived feeder cells makes scalability challenging for widespread clinical use.

aAPCs offer a more controlled and reproducible alternative to feeder cells. By expressing specific ligands and co-stimulatory molecules, aAPCs can selectively activate γδ T cells while avoiding many of the constraints associated with natural feeder cells. However, aAPCs are expensive to produce, and their complexity limits scalability, making this approach less accessible for large-scale clinical deployment [136].

Phosphoantigen-based stimulation and cytokines are a more cost-effective, simple adaptable and straightforward method. However, the variability in dosing and concentration necessitates extensive optimization for consistent results, which can increase time and resource investment during clinical translation.

With an increasing number of companies advancing γδ T cell-based or γδ T cell-engaging therapies, the substantial costs associated with manufacturing these products has been highlighted, largely driven by the need for multiple cytokines and extended ex vivo expansion processes [83]. To address these barriers, Ferry et al. introduced a streamlined one-step protocol that uses a single cytokine to efficiently expand Vδ1^+^ T cells [137]. Similarly, the Delta One T (DOT) protocol demonstrates a two-step, feeder-free approach that achieves expansions in clinical-grade Vδ1^+^ T cells by more than 2000-fold while maintaining functionality, representing another significant step toward reducing production costs and improving scalability [90].

Future improvements in automation, process standardization, and the integration of cost-efficient engineered solutions will be essential to overcoming these challenges and advancing γδ T cell therapies toward widespread clinical application.

## 7. Concluding Remarks

γδ T cells hold significant promise as agents in cancer immunotherapy due to their unique ability to recognize and eliminate tumor cells through both TCR-dependent and independent pathways. Their dual functionality, encompassing cytotoxicity and immunomodulation, positions them as versatile tools in the fight against cancer. However, realizing their full therapeutic potential necessitates the optimization of expansion protocols and the refinement of clinical application strategies.

Moving forward, continued preclinical and clinical research is essential to address the difficulties of variable patient responses and to identify the most effective combinations and engineering strategies. The integration of γδ T cells into multimodal cancer therapies could revolutionize immunotherapy by offering adaptable, potent, and safer therapeutic options. By optimizing their expansion and application, γδ T cells may pave the way for new frontiers in personalized cancer treatment.

## Figures and Tables

**Figure 1 cancers-17-01063-f001:**
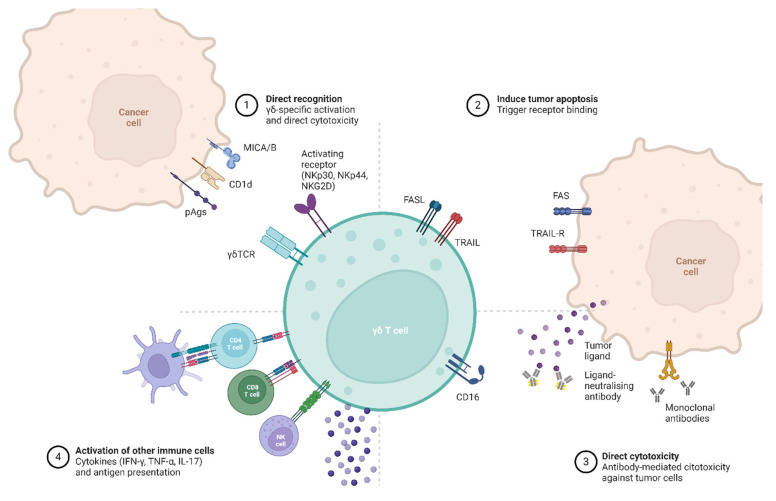
Antitumor roles of γδ T cells. Human γδ T cells exhibit potent antitumor functions upon activation through the γδ T cell receptor (TCR), natural killer receptors (NKRs), or CD16, which enables antibody-dependent cellular cytotoxicity (ADCC) by binding to antibodies attached to tumor-associated antigens or monoclonal antibodies. Tumor cell elimination is achieved either by releasing cytolytic granules filled with perforin and granzymes or by engaging death receptors using TRAIL and FASL ligands. Additionally, γδ T cells promote the cytotoxic activity of αβ T cells by releasing IFN-γ and TNF-α, which enhance MHC class I expression and antigen presentation on tumor cells. Furthermore, γδ T cells can activate NK cells via the 4-1BBL and 4-1BB pathway, strengthening the immune response against tumors and are capable of cross-presenting antigens to both CD4^+^ and CD8^+^ T cells, which helps amplify antitumor activity by recruiting and activating these additional T-cell populations. Created in BioRender. Gil, R. (2024).

**Figure 2 cancers-17-01063-f002:**
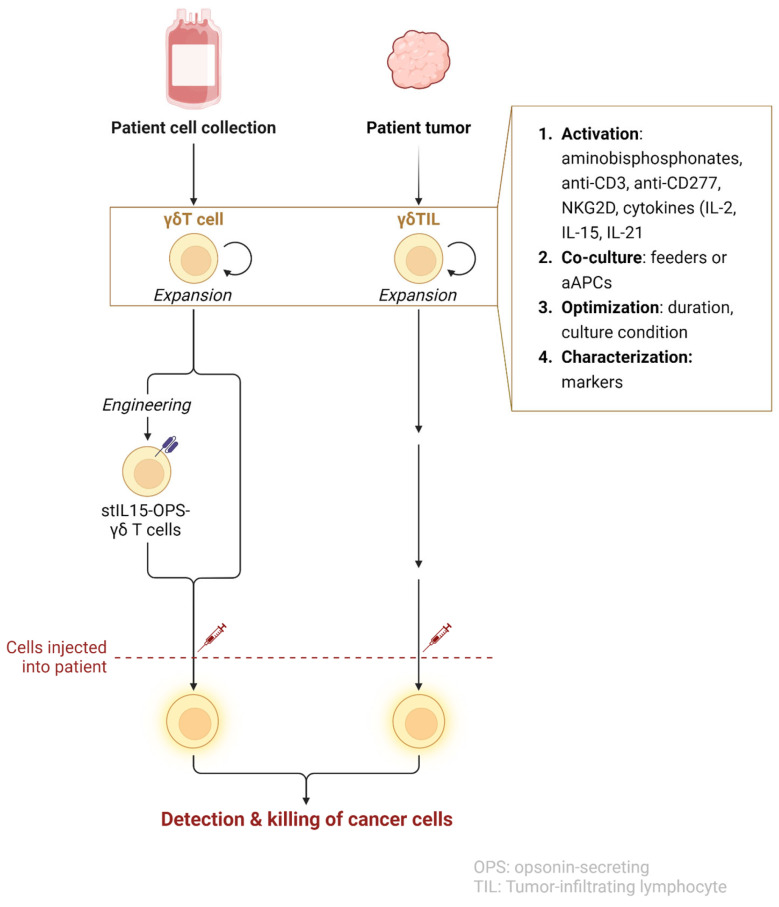
Schematic representation of the process for obtaining and preparing γδ T cells for adoptive immunotherapy. Peripheral blood or tumor samples are collected from the patient to isolate γδ T cells. These cells undergo an initial activation phase (circular arrows represent the in vitro expansion of cells prior to infusion), where they are stimulated with specific agents, such as phosphoantigens (pAgs), aminobisphosphonates, or cytokines (e.g., IL-2, IL-15), which are chosen based on desired expansion rates and phenotypic characteristics. Co-culture systems, including feeder cells or artificial antigen-presenting cells (aAPCs), are employed to support γδ T cell expansion, delivering crucial co-stimulatory signals to maintain cell viability and enhance proliferation. The optimization of culture parameters—such as oxygen concentration, cytokine levels, and duration of culture—is essential to prevent exhaustion and ensure robust, functional γδ T cells. The expanded γδ T cells are thoroughly characterized by surface markers (e.g., PD-1, NKG2D) and functional assays to confirm their cytotoxic potential, activation state, and readiness for therapeutic use. The final γδ T cell product is then prepared for infusion back into the patient as part of personalized cancer immunotherapy. Created in BioRender. Gil, R. (2024).

**Table 1 cancers-17-01063-t001:** Classification of γδ T cell subsets and comparison with αβ T cells.

Cell Subset	Vδ1	Vδ2	Vδ3	αβ T Cells
**Tissue** **Distribution**	Predominantly tissue-resident (gut epithelium, dermis, liver, spleen)	Circulating blood, some lymphoid tissues	Enriched in liver and gut; rare in blood	Found in blood, lymph nodes, and tissues based on antigen-specific responses
**Receptors**	γδ TCR (Vδ1 chain), NKG2D [22], NKp46 [23], DNAM-1 [24], CD16 [25] 2025-03-21 11:17:00	γδ TCR (Vγ9Vδ2 pairing), NKG2D [26], CD16 [27], TRAIL [28], DNAM-1 [24] 2025-03-21 11:17:00	γδ TCR (Vδ3 chain), similar to Vδ1	αβ TCR, CD4, CD8
**Antigen Recognition**	Recognizes MHC-like molecules (e.g., CD1d, CD1c, MICA/B) and lipopeptides [29,30,31,32]	Recognizes phosphoantigens in an MHC-independent manner, ULBP4, F1-ATPase [26,33]	Annexin A2 [34]	MHC-restricted recognition of peptide antigens presented by HLA
**Cytotoxic Pathways**	TRAIL, FasL, perforin–granzyme [8]	Perforin–granzyme, TRAIL, FasL ADCC via CD16 [8]	¿?	Perforin–granzyme, TRAIL, FasL

**Table 2 cancers-17-01063-t002:** Summary of completed clinical trials using γδ T cell-based therapies in cancer treatment.

Therapeutic Agent	Cancer Type	Endpoints	Reference
**Autologous γδ T Cells**	Recurrent non-small-cell lung cancer (NSCLC)	Response rate	[65]
**Autologous γδ T Cells plus** **Gemcitabine (GEM)**	Resected pancreatic cancer	Recurrence-free survival, overall survival compared to GEM alone	[66]
**Autologous γδ T cells**	Advanced non-small-cell lung cancer (NSCLC)	Median survival, median progression-free survival (PFS)	[67]
**Autologous Vγ9Vδ2 T-cells**	Refractory non-small-cell lung cancer (NSCLC)	Partial response, PFS, median overall survival (OS)	[68]
**Autologous γδ T cells alone or with chemotherapy**	Recurrent or metastatic esophageal cancer	Median OS, PFS	[69]
**Autologous Vγ9γδ T cells**	Multiple myeloma	Safety	[70]
**Autologous Vγ9Vδ2 T-cells**	Metastatic solid tumors	Dose-limiting toxicity, remission rate and disease progression	[71]
**Vγ9Vδ2 T cells plus** **zoledronate**	Malignant ascites resulting from peritoneal dissemination of gastric cancer	Safety, local antitumor effects	[72]
**Autologous γδ T cells**	Advanced renal cell carcinoma	Safety, response rate	[73]
**Innacell γδ™**	Metastatic renal cell carcinoma	Dose-limiting toxicity, response rate	[74]
**Allogeneic γδ T cells**	Refractory or relapsed acute myeloid leukemia	Safety, response rate	[76]

**Table 3 cancers-17-01063-t003:** Overview of ongoing clinical trials utilizing γδ T cell-based immunotherapies.

Trial ID	Therapeutic Agent	Targeted Cancer Types	Endpoints	Phase	Status
NCT04765462	Allogeneic Vγ9Vδ2 T Cells	Solid Tumors	Incidence of dose-limiting toxicity (DLT) and severe adverse events	Phase I/II	Ongoing
NCT04764513	Ex vivo expanded γδ T cell	Hematological Malignancies	Incidence of treatment-emergent adverse events (AEs), safety and efficacy	Phase I/II	Recruiting
NCT06404281	γδ T-PD-1 Ab cells	Advanced Solid Tumors	Incidence of dose-limiting toxicities (DLTs), incidence of adverse events (AEs)	Phase I	Recruiting
NCT04165941	DRI cell therapy (Drug Resistant Immunotherapy γδ T Cells)	Glioblastoma	Highest safe dose frequency or maximally planned dose	Phase I	Active, not recruiting
NCT03533816	EAGD T-cell infusión (Ex Vivo Expanded/Activated Gamma Delta T-cell)	Hematological Malignancies	Incidence of dose-limiting toxicities (DLTs), incidence of adverse events (AEs), Rate of acute GVHD	Phase I	Recruiting
NCT05015426	Ex vivo expanded γδ T cell	Acute Myeloid Leukemia at High Risk of Relapse	Maximum tolerated dose, leukemia free survival	Phase I/Ib	Active, not recruiting
NCT05400603	Allogeneic Ex Vivo Expanded Gamma Delta (γδ) T Cells	Relapsed/Refractory Neuroblastoma or Refractory/Relapsed Osteosarcoma	Maximum tolerated dose	Phase I	Recruiting
NCT05358808	TCB008 (Allogeneic Vγ9Vδ2 T Cells)	Refractory or Relapsed Acute Myeloid Leukaemia	Efficacy of TCB008	Phase II	Recruiting
NCT04735471	ADI-001 (Anti-CD20 CAR-engineered allogeneic γδ T Cells)	Refractory or Relapsed B cell malignancies	Safety and tolerability	Phase I	Active, not recruiting

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
