# Peer review of "γδ T Cells: Game Changers in Immune Cell Therapy for Cancer"

_cancers, 2025, doi:10.3390/cancers17071063_

Round 1

Reviewer 1 Report

Comments and Suggestions for Authors

The present manuscript entitled "γδ T Cells: Game Changers in immune cell therapy for cancer" covers clinical trial examples, illustrating real-world applications and safety profiles of γδ T cell therapies. The manuscript provides a detailed overview of γδ T cells, their roles in cancer immunotherapy, and their unique attributes compared to αβ T cells. It highlights recent advancements in γδ T cell research, expansion protocols, and clinical trials.

Here are my comments that should be addressed in the revision in order to improve the quality of the manuscript. 

1. The manuscript does not fully explore the biological complexities of γδ T cells, such as their dual roles in tumor promotion and suppression or heterogeneity within subsets, which could impact therapeutic outcomes.

2. Although challenges like cell exhaustion and persistence in the tumor microenvironment (TME) are mentioned, detailed strategies for overcoming these are inadequately explored.

3. Provide a comparative analysis of existing γδ T cell expansion protocols, including cost-effectiveness, scalability, and clinical translation hurdles.

4. The authors need to analyze trial designs in more detail, emphasizing endpoints like survival, quality of life, and biomarkers predictive of response to γδ T cell therapies.

5. The authors need to elaborate on integrating γδ T cell therapy with CAR-T cells, bispecific antibodies, or checkpoint inhibitors. Include case studies or preclinical data supporting these combinations.

6. Include Visual aids like schematics of γδ T cell mechanisms, clinical trial designs, or therapeutic strategies that could improve reader engagement and understanding.

Author Response

  1. The manuscript does not fully explore the biological complexities of γδ T cells, such as their dual roles in tumor promotion and suppression or heterogeneity within subsets, which could impact therapeutic outcomes.

We thank the reviewer for highlighting this important aspect. We have revised the manuscript to include these topics and provided additional references to support the discussion (page 5, line 196-232).

  1. Although challenges like cell exhaustion and persistence in the tumor microenvironment (TME) are mentioned, detailed strategies for overcoming these are inadequately explored.

We appreciate the reviewer’s observation regarding the need for a more detailed discussion of strategies to address cell exhaustion and persistence in the TME. In response, we have added a new section “5.4 Overcoming T cell exhaustion and enhancing persistence in the tumor microenvironment” to include an in-depth exploration of approaches to mitigate these challenges, including the use of cytokines, genetic engineering, and combination therapies (page 16, line 572-602).

  1. Provide a comparative analysis of existing γδ T cell expansion protocols, including cost-effectiveness, scalability, and clinical translation hurdles.

Thank you for your suggestion. We have expanded the manuscript to include a detailed discussion of various expansion methodologies, their relative cost-effectiveness, scalability for large-scale production, and potential challenges in clinical translation (page 17, lines 628-660).

  1. The authors need to analyze trial designs in more detail, emphasizing endpoints like survival, quality of life, and biomarkers predictive of response to γδ T cell therapies.

We thank the reviewer for pointing out this need We have expanded the section on active clinical trials to include a detailed analysis of trial endpoints (page 8, lines 298-327)

  1. The authors need to elaborate on integrating γδ T cell therapy with CAR-T cells, bispecific antibodies, or checkpoint inhibitors. Include case studies or preclinical data supporting these combinations.

We appreciate the suggestion to explore the integration of γδ T cell therapy with CAR-T cells, bispecific antibodies, and checkpoint inhibitors. We have expanded the discussion to include preclinical studies and case examples that highlight the potential synergy of these combinations (page 9, lines 345-404).

  1. Include Visual aids like schematics of γδ T cell mechanisms, clinical trial designs, or therapeutic strategies that could improve reader engagement and understanding.

Thanks for your suggestion. We included a table comparing γδ T cells with ab T cells as well as a table with past clinical trials.

Reviewer 2 Report

Comments and Suggestions for Authors

1.        Providing a bit of introduction to γδ T cells in the context of cancer immunotherapy in the abstract would be ideal before continuing to detail what the review is all about!

2.        In lines 28-31, the authors made a γδT cell comparison bet ween humans and tissue. Which tissues are the authors are referring to? Do the authors think it a feasible comparison? 

3.        While classifying the γδ T cells, the authors should consider providing a table that would help the readers to have subsets of the γδ T cells, their distribution, receptors they carry, cytokines they secrete and many as such to have a clear definition of γδ T cells. A comparison with αβ T cells will further help in seeing a side-by-side differences in one table.

4.        Similarly, before detailing the strategies for γδ T cells expansion towards the need for focusing on cancer immunotherapy, it would be ideal if the authors can present the current success rate of γδ T cells clinically. A table detailing the same will also strengthen the review as well!

5.        γδ T cells are actively recruited to the TME and have the potential to kill target cells in vitro. However, TME attenuates γδ T cell responses in multiple ways and understanding how the TME suppresses γδ T cells remains a major challenge. Can the authors expand on these lines for better understanding the protocols for in vitro expansion of γδ T cells

Comments on the Quality of English Language

Can be improved with precised wording.

Author Response

  1. Providing a bit of introduction to γδ T cells in the context of cancer immunotherapy in the abstract would be ideal before continuing to detail what the review is all about!

Thanks for the suggestion We have included a concise introduction in the abstract, highlighting the unique properties of γδ T cells (page 1, lines 11-14).

  1. In lines 28-31, the authors made a γδT cell comparison bet ween humans and tissue. Which tissues are the authors are referring to? Do the authors think it a feasible comparison?

We thank the reviewer for their comment and appreciate the opportunity to clarify and expand on this point. In this sentence, we are referring to the presence of γδ T cells in other tissues such as the gut epithelium, dermis, spleen, and liver, where γδ T cells are found in significantly higher proportions compared to circulating blood. We have updated the manuscript to specify these tissues, clarifying the confusion (lines 33-37).

  1. While classifying the γδ T cells, the authors should consider providing a table that would help the readers to have subsets of the γδ T cells, their distribution, receptors they carry, cytokines they secrete and many as such to have a clear definition of γδ T cells. A comparison with αβ T cells will further help in seeing a side-by-side differences in one table.

Thanks for this insightful suggestion. To improve clarity and facilitate understanding, we included a comprehensive table summarizing the subsets of γδ T cells, their tissue distribution, receptors, antigen recognition, and comparison to αβ T Cells (page 3, Table 1).

  1. Similarly, before detailing the strategies for γδ T cells expansion towards the need for focusing on cancer immunotherapy, it would be ideal if the authors can present the current success rate of γδ T cells clinically. A table detailing the same will also strengthen the review as well!

Thank you for pointing this out. We incorporated a new section section summarizing the clinical progress of γδ T cell-based therapies, including data from completed trials evaluating efficacy, safety, and response rates. Additionally, we provided a table highlighting key γδ T cell clinical trials in cancer, and their endpoints (page 6, lines 235-290; Table 2).

  1. γδ T cells are actively recruited to the TME and have the potential to kill target cells in vitro. However, TME attenuates γδ T cell responses in multiple ways and understanding how the TME suppresses γδ T cells remains a major challenge. Can the authors expand on these lines for better understanding the protocols for in vitro expansion of γδ T cells

Thanks for the insightful comment. We have expanded our discussion on how the TME suppresses γδ T cells and the implications for optimizing in vitro expansion protocols (page 16, lines 572-602).

Reviewer 3 Report

Comments and Suggestions for Authors

In this review article, the authors have emphasized developing and optimizing the in vitro expansion protocols, detailed activation strategies, co-culture systems, various cytokines, and other critical parameters to ensure robust cell proliferation and functionality. Finally, they discussed current progress in γδ T cell basic research and clinical development and highlighted areas needing further exploration.

Generally, it is a well-organized and presented review. However, the quality could be improved after some revisions.

There have been quite a number of reviews on the same topic in the last few years. Therefore, authors are encouraged to list a few examples of late reviews in the Introduction about them for the readers to get further ideas on this unique type of immune cells for immunotherapy of cancer. Here are a few examples,

(1).  [PMID: 38007241] Therapeutic avenues for γδ T cells in cancer. J Immunother Cancer. 2023;11:e007955. doi: 10.1136/jitc-2023-007955.

(2). [PMID: 37989744] γδ T cells: origin and fate, subsets, diseases and immunotherapy.  Signal Transduct Target Ther. 2023;8:434. doi: 10.1038/s41392-023-01653-8.

(3). [PMID: 39361750] Cancer immunotherapy by γδ T cells. Science. 2024; 386:eabq7248. doi: 10.1126/science.abq7248.

There are also some minor issues.

1.      Table 1. I would encourage authors to expand the Table 1 as quite a few completed clinical trials with gdT cells are not listed in the current version.

2.      In the database of clinicaltrials.gov, a large fraction of clinical trials with T cells have been terminated. Do you know the reasons or have some speculations?

3.      References: there is a long list of references in which essential information (mostly article number) is either missing or wrong.

Ref #11; 13; 23; 53; 74; 75; 76; 96.

Ref #16: (1). Who are the authors?  (2). The correct page numbers are: 847-59.

Ref #18: The name of the journal, year, volume and page numbers??

Ref #19. This journal has article number citable. In this case, it is 434. (not 1-38)

Ref #34. Please use the consistent format and all the essential information.

Ref #37. The correct page numbers are 433-442.

Ref #55. The correct page numbers are 411-433.

Ref #60. Correct format and essential info.

Ref #84. Correct format and essential info.

Author Response

  1. There have been quite a number of reviews on the same topic in the last few years. Therefore, authors are encouraged to list a few examples of late reviews in the Introduction about them for the readers to get further ideas on this unique type of immune cells for immunotherapy of cancer. Here are a few examples,

(1).  [PMID: 38007241] Therapeutic avenues for γδ T cells in cancer. J Immunother Cancer. 2023;11:e007955. doi: 10.1136/jitc-2023-007955.

(2). [PMID: 37989744] γδ T cells: origin and fate, subsets, diseases and immunotherapy.  Signal Transduct Target Ther. 2023;8:434. doi: 10.1038/s41392-023-01653-8.

(3). [PMID: 39361750] Cancer immunotherapy by γδ T cells. Science. 2024; 386:eabq7248. doi: 10.1126/science.abq7248.

We appreciate the reviewer’s suggestion and acknowledge the importance of providing context regarding recent literature in the field of γδ T cell-based immunotherapies. To address this, we have added references to key recent reviews in the Introduction section, highlighting their contributions while distinguishing the unique focus of our review (page 2, lines 51-64)

There are also some minor issues.

  1. Table 1. I would encourage authors to expand the Table 1 as quite a few completed clinical trials with gdT cells are not listed in the current version.

We thank your suggestion. To address this, we have added a new Table 2 to include completed clinical trials investigating γδ T cell adoptive therapies (page 7).

  1. In the database of clinicaltrials.gov, a large fraction of clinical trials with T cells have been terminated. Do you know the reasons or have some speculations?

Thank you for your insightful question While there is no single reason why these trials are discontinued, several key factors contribute to the early termination of γδ T cell-based and broader T cell immunotherapy clinical trials. Some potential reasons based on existing literature and case studies are:

- Limited clinical efficacy.

- manufacturing and scalability issues.

- Regulatory challenges.

- Competition from other immunotherapies (e.g, CAR-T)

  1. References: there is a long list of references in which essential information (mostly article number) is either missing or wrong.

Ref #11; 13; 23; 53; 74; 75; 76; 96.

Ref #16: (1). Who are the authors?  (2). The correct page numbers are: 847-59.

Ref #18: The name of the journal, year, volume and page numbers??

Ref #19. This journal has article number citable. In this case, it is 434. (not 1-38)

Ref #34. Please use the consistent format and all the essential information.

Ref #37. The correct page numbers are 433-442.

Ref #55. The correct page numbers are 411-433.

Ref #60. Correct format and essential info.

Ref #84. Correct format and essential info.

We sincerely appreciate the reviewer’s attention to detail and their request for accurate referencing. To address this concern, we have thoroughly reviewed and cross-checked all references to ensure they are correctly formatted and include all essential information (DOI numbers and article numbers where applicable).

Reviewer 4 Report

Comments and Suggestions for Authors

γδ T Cells: Game Changers in immune cell therapy for cancer

Reviewer:

This article systematically reviews the research progress of γδ T cells in cancer immunotherapy, focusing on their unique antitumor mechanisms, optimization of in vitro expansion techniques, and current clinical applications. The content covers the subset characteristics, functional differences, and potential of engineering strategies to enhance efficacy, with particular emphasis on research involving the Vγ9Vδ2 subset. In addition, the authors propose innovative hypotheses, such as expanding the therapeutic applications of γδ T cells through immune checkpoint tolerance engineering and localized cytokine delivery.

The article is well-structured and focused, offering a comprehensive perspective on the current field of γδ T cell research, particularly in the areas of expansion techniques and mechanistic understanding. However, the article has certain shortcomings, with insufficient depth in the discussion of some key areas, which affects the overall completeness and guidance value of the paper.

General Comments:

1.     While your review mentions the application of γδ T cells in certain clinical trials, I did not seem to find a systematic discussion and comparison of clinically relevant data in the article. Should this aspect be further discussed to enhance the comprehensiveness of the review?

2.     In this article, you focused on the Vγ9Vδ2 subset of γδ T cells, but the discussion on Vδ1 T cells appears to be somewhat limited. While Vγ9Vδ2 T cells are more prevalent in peripheral blood and currently the most widely studied subset in γδ T cell research, you provided a detailed analysis of their expansion techniques, antitumor mechanisms, and clinical applications. In contrast, although the progress in Vδ1 T cell research was mentioned, there was no comprehensive exploration of their expansion challenges, roles in solid tumors, and potential advantages in allogeneic therapies. Therefore, I believe that a more in-depth analysis of Vδ1 T cells could present a more balanced view of γδ T cells in tumor immunotherapy, thereby enhancing the article’s reference value.

3.     This article mentions the challenges faced by γδ T cells in the tumor microenvironment (TME), such as cell exhaustion and functional limitations. However, it seems to lack an in-depth discussion of mechanisms, for example, could the molecular mechanisms that suppress γδ T cell function in the TME be analyzed in detail? Are there any related studies? Adding a mechanistic discussion would make this article more comprehensive.

4.     Although the article mentions the potential of immune checkpoint inhibitors (e.g., PD-1 blockers) in enhancing the antitumor function of γδ T cells, the discussion is limited to a brief description. Regarding CAR-T therapy, it is only briefly mentioned in the genetic engineering section, lacking systematic analysis and supporting examples. This section would benefit from the addition of case studies to enhance the credibility of the article.

Specific Comments

1.     The logical flow of the contextual relationships in this article is somewhat problematic. For example, in the section “4. Protocols for In Vitro Expansion of γδ T Cells,” the discussion on activation strategies and co-culture systems does not naturally transition to the evaluation of cell functions (e.g., effector cytotoxicity and persistence). This makes the paragraphs appear disjointed and less coherent. Similarly, in the section “5. Clinical Trials of Gamma Delta T Cells in Cancer Immunotherapy,” the trials are listed without thoroughly discussing their relevance to the expansion techniques, making them seem isolated. Please carefully review these issues.

2.     The language needs to be refined, particularly in terms of the accuracy of technical terminology and the conciseness of sentence structure.

3.     Please double-check the citation format to ensure accuracy and consistency in the references.

Overall, the article is well-structured and logically coherent, providing a comprehensive review of the research progress of γδ T cells in cancer immunotherapy. It offers rich information and theoretical foundations on in vitro expansion techniques, mechanistic understanding of the Vγ9Vδ2 subset, and engineering modifications, while also proposing several innovative hypotheses with a forward-looking perspective. However, there is room for improvement in the depth and breadth of content. A major revision is recommended, with a focus on addressing the key issues to enhance the academic value of this article.

Author Response

  1. While your review mentions the application of γδ T cells in certain clinical trials, I did not seem to find a systematic discussion and comparison of clinically relevant data in the article. Should this aspect be further discussed to enhance the comprehensiveness of the review?

We thank the reviewer for highlighting this need. We have added a summary of key aspects of these studies—including trial design, patient cohorts, endpoints, outcomes, and safety profiles—to offer readers a clearer perspective on the current clinical landscape of γδ T cell-based immunotherapies (page 6, lines 234-287).

  1. In this article, you focused on the Vγ9Vδ2 subset of γδ T cells, but the discussion on Vδ1 T cells appears to be somewhat limited. While Vγ9Vδ2 T cells are more prevalent in peripheral blood and currently the most widely studied subset in γδ T cell research, you provided a detailed analysis of their expansion techniques, antitumor mechanisms, and clinical applications. In contrast, although the progress in Vδ1 T cell research was mentioned, there was no comprehensive exploration of their expansion challenges, roles in solid tumors, and potential advantages in allogeneic therapies. Therefore, I believe that a more in-depth analysis of Vδ1 T cells could present a more balanced view of γδ T cells in tumor immunotherapy, thereby enhancing the article’s reference value.

Thank you for your valuable feedback. To address this, we expanded the discussion on Vδ1 T cells (page4 lines 148-157; page 8, lines 267-271; page 9, lines 345-349; page 11, lines 411-419)

3. This article mentions the challenges faced by γδ T cells in the tumor microenvironment (TME), such as cell exhaustion and functional limitations However, it seems to lack an in-depth discussion of mechanisms, for example, could the molecular mechanisms that suppress γδ T cell function in the TME be analyzed in detail? Are there any related studies? Adding a mechanistic discussion would make this article more comprehensive.

We appreciate the suggestion. We have deepened our discussion on the strategies to overcome the limitations in the TME providing a detailed analysis (page 16, lines 577-605).

4. Although the article mentions the potential of immune checkpoint inhibitors (e.g., PD-1 blockers) in enhancing the antitumor function of γδ T cells, the discussion is limited to a brief description. Regarding CAR-T therapy, it is only briefly mentioned in the genetic engineering section, lacking systematic analysis and supporting examples. This section would benefit from the addition of case studies to enhance the credibility of the article.

Thanks for your suggestion. We have substantially revised and enriched the section (page 9, lines 353-410)

Specific Comments

  1. The logical flow of the contextual relationships in this article is somewhat problematic. For example, in the section “4. Protocols for In Vitro Expansion of γδ T Cells,” the discussion on activation strategies and co-culture systems does not naturally transition to the evaluation of cell functions (e.g., effector cytotoxicity and persistence). This makes the paragraphs appear disjointed and less coherent. Similarly, in the section “5. Clinical Trials of Gamma Delta T Cells in Cancer Immunotherapy,” the trials are listed without thoroughly discussing their relevance to the expansion techniques, making them seem isolated. Please carefully review these issues.

We appreciate the reviewer’s comments. We have carefully restructured key sections to improve coherence and ensure a more natural transition between topics.

  1. Overview of γδ T Cells in the Immune System
  2. Differences Between γδ T Cells and αβ T Cells
  3. Classification and Anti-Tumor Functions of γδ T Cells
  4. γδ T Cell-Based Immunotherapies: Clinical Applications and Challenges

4.1 Clinical Outcomes of γδ T Cell-Based Therapies

4.2 Current Clinical Trials and Combination Strategies

  1. Optimization of γδ T Cell Expansion for Therapeutic Applications
    • Activation and Stimulation Approaches
    • 2 Co-Culture Systems and Feeder Cells
    • Expansion Protocol Optimization
    • Strategies for Overcoming Exhaustion and Enhancing Persistence in the Tumor Microenvironment
  2. Comparative Analysis and Characterization of γδ T Cell Expansion Protocols
  3. Concluding Remarks

  1. The language needs to be refined, particularly in terms of the accuracy of technical terminology and the conciseness of sentence structure.

We appreciate the reviewer’s feedback regarding language refinement. We have carefully revised the manuscript to improve technical accuracy, clarity, and conciseness.

  1. Please double-check the citation format to ensure accuracy and consistency in the references.

We appreciate the reviewer’s attention to detail and their request for accurate referencing. To address this concern, we have thoroughly reviewed and cross-checked all references to ensure they are correctly formatted and include all essential information.

Round 2

Reviewer 1 Report

Comments and Suggestions for Authors

The authors have responded to my comments in the revised manuscript.

Reviewer 4 Report

Comments and Suggestions for Authors

This study provides a comprehensive review of the biological properties, expansion strategies, clinical applications, and prospects of γδ T cells in cancer immunotherapy. The article is well-structured, supported by an extensive range of literature references, and offers a thorough discussion of key topics in the field. Compared to the initial draft, the authors have made significant improvements. First, they supplemented the discussion with clinical trial data on γδ T cells, enhancing the completeness of the literature review. Second, they expanded the analysis of Vδ1 T cells, strengthened the mechanistic discussion on the impact of the tumor microenvironment (TME) on γδ T cell function, and reorganized sections to improve logical coherence.

Overall, these revisions have substantially enhanced the academic rigor and structural clarity of the article. However, a few minor issues remain that require attention, which, if addressed, will further refine the manuscript and strengthen its impact.

Minor Comments:

  1. The article mentions several clinical trials on γδ T cells, but does not provide an in-depth comparison of their results. A comparison of key trial data side-by-side would improve the discussion. In particular, it would be good to mention: 1. In terms of objective remission rate (ORR) and overall survival (OS), how did γδ T cells compare across different trials? A comparative table summing up these results would illustrate their clinical efficacy more clearly. 2. Is there a difference in the effectiveness of γδ T cell therapy across cancers? If so, why? Discussing factors such as tumor microenvironment interactions or antigen expression levels would strengthen the analysis. 3. How does γδ T cell therapy compare to other immunotherapies like CAR-T or PD-1 inhibitors? Emphasizing the unique benefits and drawbacks of γδ T cells compared to these established approaches would provide a more even analysis. These comparisons would improve the clinical relevance and practical applications of the review. Since there is much content already in this article, the above suggestions are for reference only and at the author’s discretion.
  2. Your paper is overall very wordy, so I suggest taking a look at the overall logical flow to ensure clarity. Please consider the discussion section consisting of five topics in particular. There are areas that could be improved—the logical flow between these five topics is very weak, and it seems more like you are ending one topic and beginning another rather than continuing discussion. Improving these transitions would enable better readability and overall impact of the discussion.
  3. There are still some inconsistencies in the formatting of references; please review and ensure uniformity.
  4. I noticed that the term CAR-T was not spelled out in full when it first appeared. Please check and provide the full term upon its first mention.
  5. The formatting of tables could be improved for better readability. Consider bolding key data to highlight important information.

This review article provides a comprehensive analysis of γδ T cells in cancer immunotherapy, covering their biological characteristics, expansion strategies, clinical applications, and future directions. The revised version has significantly improved in content completeness, structural clarity, and academic rigor, particularly by expanding discussions on clinical trials, Vδ1 T cells, and tumor microenvironment interactions. However, some logical inconsistencies remain, especially in the discussion section’s topic transitions. Further refinement of clinical trial data analysis, comparisons with other immunotherapies, and table formatting is recommended. Minor issues, such as reference inconsistencies and terminology clarity, should also be addressed. Overall, minor revisions are needed before publication.

Comments on the Quality of English Language

Your paper is overall very wordy, so I suggest taking a look at the overall logical flow to ensure clarity. Please consider the discussion section consisting of five topics in particular. There are areas that could be improved—the logical flow between these five topics is very weak, and it seems more like you are ending one topic and beginning another rather than continuing discussion. Improving these transitions would enable better readability and overall impact of the discussion.